# Gestational Diabetes Mellitus Subtypes Derived by Clustering Analysis Show Heterogeneity in Glucometabolic Parameters Already at Early Pregnancy

**DOI:** 10.3390/nu17203252

**Published:** 2025-10-16

**Authors:** Grammata Kotzaeridi, Benedetta Salvatori, Agnese Piersanti, Florian Heinzl, Sophie Zarotti, Herbert Kiss, Silke Wegener, Iris Dressler-Steinbach, Wolfgang Henrich, Micaela Morettini, Andrea Tura, Christian S. Göbl

**Affiliations:** 1Department of Obstetrics and Gynecology, Medical University of Vienna, 1090 Vienna, Austria; grammata.kotzaeridi@meduniwien.ac.at (G.K.); florian.heinzl@meduniwien.ac.at (F.H.); sophie.zarotti@meduniwien.ac.at (S.Z.); herbert.kiss@meduniwien.ac.at (H.K.); 2Unit of Medical Genetics, IRCCS Istituto Giannina Gaslini, 16147 Genoa, Italy; benedetta.salvatori@in.cnr.it; 3National Research Council (CNR) Institute of Neuroscience, 35127 Padova, Italy; agnese.piersanti@in.cnr.it (A.P.); andrea.tura@cnr.it (A.T.); 4Department of Obstetrics, Charité—Universitätsmedizin Berlin, 10117 Berlin, Germany; silke.wegener@charite.de (S.W.); iris.dressler-steinbach@charite.de (I.D.-S.); wolfgang.henrich@charite.de (W.H.); 5Department of Information Engineering, Università Politecnica delle Marche, 60131 Ancona, Italy; m.morettini@staff.univpm.it

**Keywords:** GDM subtypes, early pregnancy, hyperglycemia, insulin resistance, beta-cell function, cluster analysis, web application, nutrition therapy

## Abstract

**Background/Objectives:** The classification of patients with diabetes into phenotypes with distinct risks and therapeutic needs is crucial for individualized care. We recently introduced a clustering model for gestational diabetes mellitus (GDM). This study aims to further characterize the proposed clusters and to identify cluster-specific differences in glucometabolic parameters during early pregnancy in an independent cohort. The metabolic profiles and dietary habits of GDM clusters will be compared with those of a normal glucose-tolerant (NGT) control group. **Methods:** 1088 women (195 who developed GDM and 893 who remained NGT) underwent a broad risk evaluation at early pregnancy. GDM patients were further categorized into the three proposed GDM subtypes (CL1 to CL3). **Results:** Among GDM patients, 7.7% were classified as CL1, 35.9% as CL2, and 56.4% as CL3. CL1 showed higher age, pregestational BMI, and increased glucose concentrations both at fasting and during the diagnostic oral glucose tolerance test. CL2 was characterized by elevated BMI and fasting glucose, while CL3 showed higher glucose concentrations after the oral glucose load, with BMI levels comparable to NGT mothers. Women in the CL1 group exhibited impaired insulin sensitivity and β-cell function at early pregnancy and showed elevated lipid levels. Compared to NGT women, a positive family history of diabetes was more prevalent in CL1 and CL3, but not in CL2. Dietary patterns were similar across all groups. **Conclusions**: Our study showed distinct alterations in glucometabolic parameters already at early pregnancy among GDM subtypes. Patients in CL1 exhibited the most unfavorable risk constellation and could benefit from lifestyle changes and nutrition therapy in early pregnancy, despite showing similar dietary patterns as the NGT group.

## 1. Introduction

Gestational Diabetes Mellitus (GDM) presents a complex challenge in clinical management due to its phenotypic heterogeneity. This heterogeneity reflects not only the diverse physiological characteristics arising from the underlying pathophysiological mechanism [1] but also the genetic background of affected women [2], similar to what is observed in type 2 diabetes (T2D) [3,4]. This variability necessitates a tailored approach to treatment, recognizing that while some women may achieve glycemic control through nutrition therapy and lifestyle modifications alone, others require pharmacological interventions comprising one or more types of insulin, sometimes combined with metformin [5]. Despite these efforts, some patients may still experience adverse pregnancy outcomes, such as the birth of large-for-gestational-age (LGA) infants, indicating that optimal glycemic control does not uniformly prevent all potential complications of GDM. The HAPO study and the HAPO Follow-up study showed a graded association between maternal blood glucose levels and short-term perinatal complications as well as long-term risks in both mothers and offspring [6,7], while Farrar et al. provided evidence that different levels and patterns of hyperglycemia are associated with varying risks of adverse perinatal outcomes [8]. Moreover, Catalano et al. identified maternal obesity as a key factor in adverse outcomes for both mother and child during pregnancy and beyond, with these risks further compounded by the presence of GDM [9,10].

Consequently, enhancing our understanding and differentiation of clinical phenotypes of GDM is a well-known research gap in order to provide more personalized treatment strategies based on nutrition therapy and pharmacological intervention [11].

Previous observations indicated that the heterogeneity of GDM may be explained by specific maternal characteristics, such as pregestational body mass index (BMI) [10,12,13,14,15], fasting glucose concentrations and elevated glucose levels during the 75 g oral glucose tolerance test (OGTT), as suggested previously by us and others [15,16,17,18,19], glucometabolic indices [1,20] or maternal age [15,21]. By analyzing patterns in these routinely available key variables, we have recently identified three “clusters” (or subtypes) of GDM, which showed distinct differences in patient characteristics and pregnancy outcomes [5].

The objective of the present study is to further characterize the proposed subtypes of GDM in an independent cohort of pregnant women. More specifically, we aim to assess cluster-specific differences in glucometabolic features at an early gestational age, as well as in comparison to women who maintained normal glucose tolerance (NGT) throughout pregnancy. Additionally, we aimed to compare the cluster model with a simpler classification based on isolated fasting and/or post-load hyperglycemia by measuring their concordance.

## 2. Materials and Methods

### 2.1. Study Design and Participants

A detailed description of this prospective cohort study is reported elsewhere [17,22]. In short, we prospectively included 1132 women at a median gestational age of 12.9 weeks (IQR: 1.3) visiting the outpatient clinic of our institution (Department of Obstetrics and Gynecology, Medical University of Vienna), between 2016 and 2019. Study participants received a broad risk evaluation, including the detailed assessment of obstetric history, BMI, family history and ethnicity, as well as a blood examination including fasting glucose, insulin and C-peptide, glycated hemoglobin A1c (HbA1c) and serum lipids. Maternal dietary habits at early pregnancy were assessed using a food frequency questionnaire (FFQ) administered at the time of inclusion [23]. Thereafter, participating women were followed until delivery to assess the presence of pregnancy complications. Thereby, GDM was diagnosed by a 75 g OGTT according to recommended diagnostic criteria [24]. Out of the total cohort, we excluded 44 patients due to incomplete or missing OGTT data. Therefore, the effective sample size of this report consisted of 893 NGT women and 195 GDM patients. Women with diabetes before pregnancy (such as type 1 or type 2 diabetes), as well as pregnant women with a history of bariatric surgery (such as sleeve gastrectomy or gastric bypass), were excluded from this study.

The study was approved by the Ethics Committee of the Medical University of Vienna and performed in accordance with the Declaration of Helsinki. All participants gave written informed consent.

### 2.2. Laboratory Methods and Calculations

Laboratory parameters were measured according to the standard methods (http://www.kimcl.at, accessed on 25 May 2024). International growth curves were used to assess international growth percentiles for singleton pregnancies according to [25]. Large for gestational age (LGA) was defined as newborn bodyweight above the 90th percentile.

Insulin sensitivity was assessed by the quantitative insulin sensitivity check index (QUICKI) [26] and the triglyceride-glucose insulin sensitivity index (TyGIS), which is a recently developed surrogate index for the insulin sensitivity reference standard (clamp-based M parameter) [27]. Thereafter, we assessed pancreatic β-cell function. This was accomplished by the insulinogenic index, IGI (precisely, a modified version of IGI), which was calculated from fasting parameters, as IGI_F_ = FCP [ng/mL]/FPG [mg/dl], whereby FCP is fasting C-peptide and FPG is fasting plasma glucose [28]. Finally, we calculated the oral disposition index, DI_ORAL_, which is a parameter of β-cell function adjusted for the prevailing insulin sensitivity level. The formulation of the oral disposition index that we used was DI_ORAL_ = IGI_F_ × FI^−1^, where FI is fasting insulin [29,30].

### 2.3. Classification of GDM Subgroups and Related Web Application

Women with GDM were classified into three GDM clusters (CL1 to 3) as described in detail in our previous publication [5]. Briefly, we summarize here the main aspects of our cluster model previously developed: The final clustering solution was generated using the k-means algorithm with k = 3 clusters, based on five input features: age, pregestational BMI (BMIPG), and glycemia at fasting and during the diagnostic OGTT (OGTT0, OGTT60, and OGTT120, respectively). These configurations were selected after systematically evaluating multiple clustering methods and distance metrics and validating them against pre-defined internal criteria. The quality of the identified cluster solution was finally tested according to different criteria that proved the three identified clusters to be robust, especially based on the strategy of division of the whole dataset into training and test subsets (70% and 30%, respectively). For the cluster assignment in the present study, we used a user-friendly web application (https://clugdm.shinyapps.io/clugdm/, accessed on 31 October 2024) [31].

### 2.4. Statistical Analysis

Continuous variables were summarized by mean ± standard deviation or, in the case of skewed distribution, as median and interquartile ranges (IQR), and were compared using analysis of variance (ANOVA) or rank-based inference in the case of skewed distribution [32]. The R-squared statistic was used to assess the fit of linear models. Categorical variables were summarized by counts (and percentages), and group-specific differences were assessed by use of binomial logistic regression to calculate odds ratios (OR) and corresponding 95% confidence intervals (95%CI). Tukey’s HSD was used for all subgroup (k = 4) comparisons to achieve a 95% coverage probability. The agreement of the cluster model with a simpler categorization based on either elevated fasting and/or postprandial glucose concentrations was assessed by weighted (Fleiss–Cohen) kappa statistics. Statistical analysis was performed with R (version 4.3.2) and contributing packages [33]. A two-sided *p*-value of <0.05 was considered statistically significant. All *p*-values are interpreted in an exploratory manner, aiming to generate new hypotheses. Apart from Tukey’s HSD for the adjustment of the comparison of multiple groups in a specific parameter, no further adjustment for multiple testing was used.

### 2.5. Sample Size Justification

As this is a secondary analysis of a previous study, we performed no post hoc power calculation. However, the sample size of the original study, aiming to assess the predictive value of several prediction models for GDM development, was justified in our previous publication [22].

## 3. Results

### 3.1. Characteristics of the Study Cohort

Table 1 shows characteristics of the study population, consisting of 893 NGT women and 195 GDM patients. Among GDM patients, 15 (7.7%) were classified as CL1, 70 (35.9%) as CL2, and 110 (56.4%) as CL3. As expected, patients in CL1 had higher age, pregestational BMI, and increased glucose concentrations during the diagnostic OGTT, both in fasting and post-load levels. Patients in CL2 were characterized by elevated BMI and fasting glucose at the diagnostic OGTT, although these parameters were lower compared to patients in CL1. In contrast, patients in CL3 showed higher glucose concentrations after the oral glucose load, while their pregestational BMI was significantly lower compared to patients in CL1 and CL2 but comparable to mothers with NGT. No significant differences in maternal dietary habits at early pregnancy were observed between NGT women and those in the different clusters, as presented in Appendix A.

### 3.2. Metabolic Assessments

The GDM subtypes exhibited distinct alterations in early gestational glucometabolic profiles, as presented in Figure 1 and Figure 2. Notably, women in CL1 already exhibited hyperglycemia (in terms of fasting glucose and HbA1c) at the start of pregnancy, which was associated with higher insulin resistance and impaired β-cell function (indicated by a lower oral disposition index, DI_ORAL_). Additionally, CL1 had an unfavorable risk profile in terms of serum lipids, with increased triglycerides and lower HDL-cholesterol levels. We also observed significant differences in family history of diabetes, which was more common in CL1 (OR: 6.35, 95%CI: 1.58 to 25.5, *p* = 0.004 vs. NGT for first degree relatives) and CL3 (OR: 2.28, 95%CI: 1.35 to 3.85, *p* < 0.001 vs. NGT), but not in CL2 (OR: 0.94, 95%CI: 0.45 to 1.97, *p* = 0.996 vs. NGT).

### 3.3. Glucose-Lowering Medication and Pregnancy Outcomes

Due to the sample size in some subtypes (especially CL1) in this study, we were unable to identify differences in binary outcomes such as pharmacotherapy (CL1: 53.3%, CL2: 44.3%, CL3: 37.2%, *p* = 0.392) or the development of LGA offspring (NGT: 14.2%, CL1: 20.0%, CL2: 20.3%, CL3: 16.7%, *p* = 0.550).

### 3.4. Comparison with a Classification Based on Isolated and Combined Fasting and Post-Prandial Hyperglycemia

CL1 contained 14 (31.8%) patients with combined hyperglycemia, 1 (1.5%) patient with isolated fasting hyperglycemia, and no patient with isolated post-load hyperglycemia. CL2 contained 4 (9.1%) patients with combined hyperglycemia, 60 (86.9%) patients with isolated fasting hyperglycemia, and 5 (6.0%) patients with isolated post-load hyperglycemia. CL3 contained 26 (59.0%) patients with combined hyperglycemia, 6 (9.0%) patients with isolated fasting hyperglycemia, and 78 (94.0%) patients with isolated post-load hyperglycemia. The weighted kappa was 0.44 (95% CI 0.30–0.58), indicating only fair to moderate agreement rather than strong concordance. When both methods of categorization were compared for their ability to identify differences in insulin sensitivity, it was found that the cluster model accounted for 10.6% of the variation, while the simple categorization based on fasting and post-load hyperglycemia accounted for 8.4% of the variation in terms of QUICKI. Additionally, when both methods were included in a single two-way ANOVA, the cluster approach remained significant (*p* = 0.007), whereas the simple hyperglycemia-based approach did not (*p* = 0.063).

## 4. Discussion

This study aimed to assess early glucometabolic alterations in recently proposed GDM subtypes as defined by clustering analysis. In this independent study, all three clusters showed worse metabolic profiles as compared to a normal glucose-tolerant control group, but with distinct alterations in glucometabolic parameters. Notably, patients in cluster 1 exhibited a higher degree of hyperglycemia, as well as insulin resistance and beta-cell dysfunction. Consequently, our study confirmed the increased risk associated with this specific subgroup. However, no statistically significant differences were observed between the clusters for binary outcomes such as pharmacotherapy use or the occurrence of LGA infants.

In a previous study, Powe et al. employed a “physiologically-based” approach to classify GDM subtypes based on underlying pathophysiological mechanisms [1]. Thereby, the authors identified 67 cases of GDM, which were further classified into three physiologic subtypes by use of dynamic indices of glucose metabolism: i. patients with predominant insulin sensitivity defect (50.7%), ii. patients with predominant secretion defect (29.9%) and iii. patients with mixed defects (17.9%). A similar approach was used in a secondary analysis of the DALI (Vitamin D and lifestyle intervention for gestational diabetes mellitus prevention) study in 236 obese mothers with “early” GDM, diagnosed before 20 weeks of gestation [34], as well as in the Belgian Diabetes in Pregnancy study, which classified GDM subtypes based on insulin sensitivity [35]. All these studies indicated that a more insulin-resistant subtype of GDM was associated with adverse pregnancy outcomes. While a “physiologically based” classification, which was used in these previous studies, is essential for uncovering the well-known heterogeneity of GDM in experimental medical research, it is important to note that categorizing GDM subgroups based on physiological traits often relies on parameters that are not routinely available in clinical practice. For instance, frequent measurements of insulin and C-peptide during the OGTT may be necessary to provide a full picture of glucometabolic alterations, although these parameters are not part of standard clinical procedures. Therefore, we recently suggested a different approach to define subgroups of GDM by use of cluster analysis, an unsupervised machine learning approach, which was also previously used to classify patients with diabetes outside of pregnancy [36]. In that earlier study [5], we examined datasets from over 2500 GDM patients treated in two central European hospitals, aiming to categorize patients according to simple and routinely available clinical parameters, which were identified as specific risk markers for GDM in previous studies (such as maternal age, pregestational BMI, and OGTT glucose levels at fasting and after oral glucose load). Through this analysis, we identified three subtypes (clusters) of the disease. These subtypes underwent subsequent validation using both internal and external procedures. Notably, the hence identified subtypes exhibited significant differences in GDM-associated complications, including the prescription of glucose-lowering medications or the development of LGA offspring, with cluster 1 showing particularly unfavorable outcomes. When we applied the cluster model in the current study, we found a similar distribution of input variables within the clusters to that observed in the original study. This alignment with the original publication further corroborates the robustness of our cluster model in general. Nevertheless, our current study has enhanced our understanding of specific characteristics associated with various GDM subtypes. Notably, patients in cluster 1 exhibited the lowest prevalence but the most unfavorable risk profile already early in pregnancy and were characterized by elevated lipids in addition to hyperglycemia, as well as beta-cell dysfunction and particularly impaired insulin sensitivity. These findings are in line with those studies using a “physiologically-based” classification approach, highlighting more adverse perinatal outcomes, particularly in GDM patients with predominant defects in insulin action as mentioned above [1,34,35]. However, in contrast to the original publication, we were unable to identify group-specific differences in pregnancy outcomes for patients in cluster 1 in our current study. This failure to demonstrate predictive value for clinical outcomes may be attributable to the small number of patients in this subgroup. Nevertheless, early gestational characteristics of patients in CL1 may hint towards an elevated risk for fetal overgrowth as previously observed for this subgroup [37,38].

Patients in cluster 2 showed elevated BMI and fasting glucose at early pregnancy, although both parameters were significantly lower than those in patients in cluster 1. Of note, pregestational BMI in cluster 3 was comparable to normal glucose-tolerant women. These observations are in line with our previous publication. Consequently, these results may be regarded as a validation of our cluster model, in general.

We also compared a simple approach that categorizes GDM patients based on isolated or combined fasting and post-load hyperglycemia solely [17], with the cluster model approach. Our analysis showed that the simple approach demonstrated only moderate to fair concordance with the cluster model, reflecting the inclusion of additional variables such as age and BMI. Additionally, the cluster model exhibited a stronger association with fasting insulin sensitivity, as indicated by the findings of the current study.

Lifestyle modification during early pregnancy, such as nutrition therapy and physical activity, may lower the risk of developing GDM and mitigate GDM-related complications. Specifically, women in CL1 with their high-risk profile—characterized by older age, higher BMI, family history of diabetes, and adverse metabolic profiles in early gestation—may derive the greatest benefit from early, targeted lifestyle interventions. The effectiveness of targeted lifestyle interventions during early pregnancy in preventing GDM has been demonstrated in a recent systematic review and meta-analysis by Takele et al. [39]. In this analysis, maternal dietary habits assessed in early pregnancy did not differ significantly between NGT women and those in various clusters. However, a potential effect of early dietary modifications cannot be excluded, given the limited sample size, especially in the CL1 group. Furthermore, a single FFQ assessment can be problematic, as it depends on retrospective self-reporting of dietary intake over the previous four weeks and may fail to capture recent or short-term dietary changes. It is possible that some women adopt less healthy dietary habits as pregnancy progresses, which can only be detected through repeated dietary assessments over time.

It is also worth mentioning that women classified into cluster 1 and cluster 3 exhibited a significantly higher proportion of family members with diabetes, a pattern not observed among patients in cluster 2. This finding may suggest the presence of unique genetic influences that contribute to the heterogeneity of GDM, as previously discussed in more detail elsewhere [2]. Indeed, Hayes et al. identified genetic loci specifically associated with elevated fasting C-peptide (a surrogate parameter of impaired insulin action) and increased postprandial glucose in pregnancy, which are predominant characteristics of clusters 1 and 3 [40]. Another recent study demonstrated that genetic variants in two loci, GCKR and MTNR1B, were associated with insulin-resistant but not insulin-deficient GDM, suggesting that such differences may shape the maternal metabolome and ultimately influence pregnancy outcomes [41]. In a large genome-wide association study (GWAS) of 12,332 GDM cases and 131,109 parous female controls, Elliott et al. identified nine novel loci associated with GDM and indicated that the genetic architecture of GDM risk is partly shared with T2D and partly defined by pregnancy-specific contributors [42]. Similarly, Powe et al. identified physiologically grouped genetic variants associated with GDM, showing that some clusters overlap with T2D mechanisms such as β-cell dysfunction and hepatic lipid metabolism, while a pregnancy-specific cluster was more strongly linked to GDM and nominally associated with higher infant birthweight [43]. However, additional research may be required to further elucidate the genotypic heterogeneity of GDM and the proposed GDM subtypes.

This study has some limitations that should be acknowledged. As this was a secondary analysis with a single-center design, the generalizability of our findings may be limited. While it remains an interesting observation that the prevalence of GDM classified as cluster 1 (CL1) was rather low, the small number of participants in this cluster may have restricted the statistical power to adequately identify cluster-specific differences in binary outcomes (such as insulin treatment or LGA rates). Given that the study population is representative of a Central European cohort, further studies with more ethnically diverse populations are needed to assess whether the results are applicable in an international context. Another limitation is the single assessment of metabolic parameters, which may not capture changes over the course of pregnancy. Furthermore, the assessment of baseline dietary intake via self-reported FFQ may have limited reliability and does not account for changes in dietary habits throughout pregnancy, which need to be addressed in future studies. Although group-based comparisons were adjusted using Tukey’s HSD, the possibility of type I error from multiple testing should be acknowledged for this exploratory study, which basically aimed to create new hypotheses.

Lastly, this study demonstrates that cluster-specific metabolic alterations are already detectable in early pregnancy, raising the prospect that cluster-based risk stratification paired with early, individualized GDM treatment strategies could possibly improve clinical outcomes. Nevertheless, well-designed interventional trials are essential to establish whether this approach indeed results in improved clinical outcomes.

## 5. Conclusions

In summary, the current study confirmed distinct differences among GDM subtypes, consistent with previous data-driven cluster analysis. Consequently, the results of this study corroborate and further validate our previously defined cluster model. Heterogeneity was already observed at early pregnancy, whereby patients in CL1 exhibited the most unfavorable risk constellation with elevated glucose and lipid concentrations as well as impaired insulin sensitivity and beta-cell function. No significant differences in binary outcomes between clusters were observed, most likely due to the small size of CL1. Early risk stratification, along with tailored lifestyle modifications such as nutrition therapy, may help reduce the risk of developing GDM and improve clinical outcomes.

## Figures and Tables

**Figure 1 nutrients-17-03252-f001:**
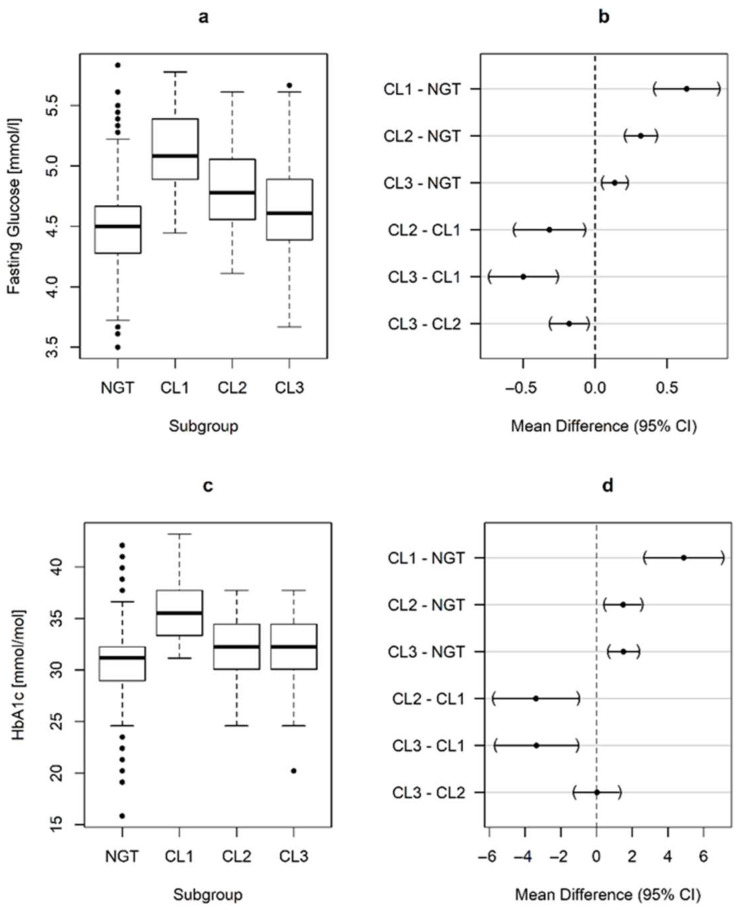
Box-whisker plots as well as estimated mean difference, and 95% family-wise confidence levels representing group-specific differences in glycemic parameters at early pregnancy. Fasting glucose (**a**,**b**), HbA1c (**c**,**d**); Normal glucose tolerant (NGT), GDM Clusters (CL).

**Figure 2 nutrients-17-03252-f002:**
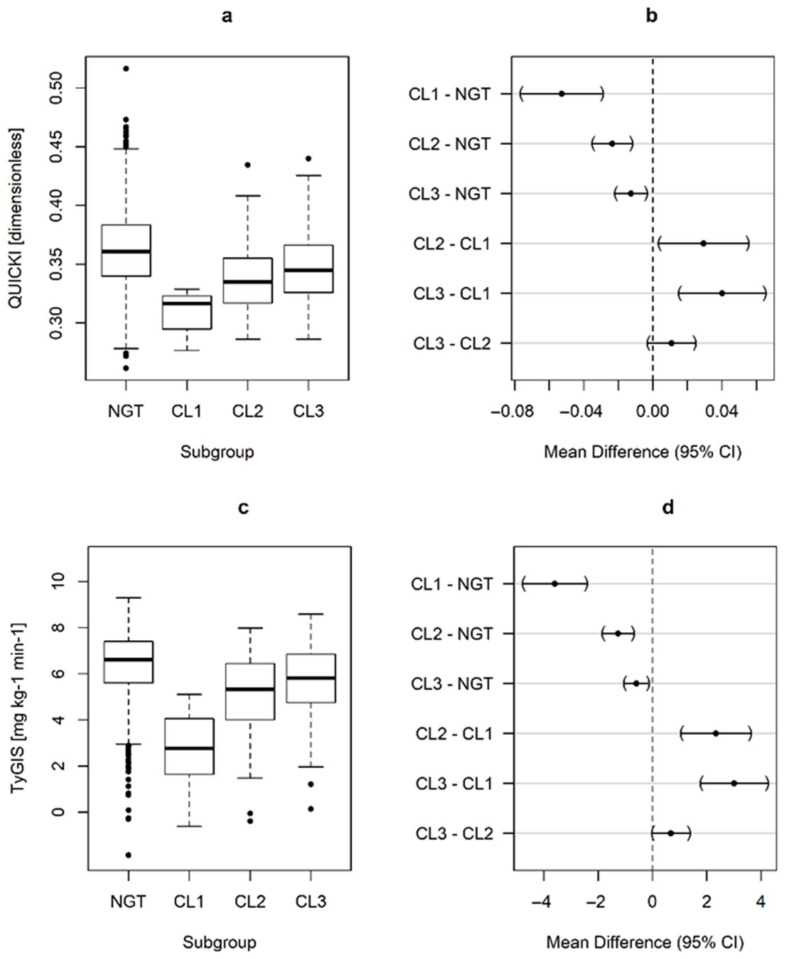
Box-whisker plots as well as estimated mean differences, and 95% family-wise confidence levels representing group-specific differences in insulin resistance parameters at early pregnancy. QUICKI (**a**,**b**), TyGIS (**c**,**d**); Normal glucose tolerant (NGT), GDM Clusters (CL).

**Table 1 nutrients-17-03252-t001:** Characteristics of the study population.

	NGT	CL1	CL2	CL3
	(n = 893)	(n = 15)	(n = 70)	(n = 110)
Age (years)	31.4 ± 5.8	35.9 ± 5.3 *	31.4 ± 5.2†	33.2 ± 5.6 *
Parity (≥1)	541 (60.6)	11 (73.3)	51 (72.9)	72 (65.5)
GDM in previous pregnancy	52 (5.8)	7 (46.7) *	14 (20.0) *	30 (27.3) *
Ethnicity (non-Caucasian)	184 (20.6)	5 (33.3)	20 (28.6)	32 (29.1)
BMI, before pregnancy (kg/m^2^)	24.3 ± 5.2	34.0 ± 5.8 *	28.3 ± 5.4 *†	25.1 ± 4.5 †§
BMI, early pregnancy (kg/m^2^)	24.8± 5.1	34.3 ± 5.5 *	29.1 ± 5.5 *†	25.8 ± 4.5 †§
Family history of diabetes (1st grade)	214 (23.9)	10 (66.7) *	16 (22.9) †	46 (41.8) *§
Family history of diabetes (1st & 2nd grade)	386 (43.2)	12 (80.0) *	33 (47.1)	74 (67.3) *§
Multiple pregnancy	107 (12.0)	0 (0.0)	8 (11.4)	10 (9.1)
Triglycerides, early pregnancy (mg/dl)	114 ± 44	158 ± 38 *	126 ± 45	139 ± 53 *
Total-cholesterol, early pregnancy (mg/dl)	188 ± 35.0	185 ± 28	185 ± 35	194 ± 37
LDL-cholesterol, early pregnancy (mg/dl)	94.1 ± 27.9	92.7 ± 26.1	96.8 ± 29.2	97.2 ± 28.0
HDL-cholesterol, early pregnancy (mg/dl)	71.1 ± 16.1	60.1 ± 12.0 *	63.6 ± 12.7 *	69.2 ± 16.2
FPG, early pregnancy (mmol/L)	4.48 ± 0.32	5.12 ± 0.44 *	4.80 ± 0.31 *†	4.62 ± 0.38 *†§
HbA1c, early pregnancy (%)	4.95 ± 0.29	5.40 ± 0.29 *	5.09 ± 0.25 *†	5.09 ± 0.30 *†
HbA1c, early pregnancy (mmol/mol)	30.6 ± 3.2	35.5 ± 3.1 *	32.1 ± 2.8 *†	32.2 ± 3.2 *†
OGTT-G 0′ (mmol/L)	4.37 ± 0.37	5.84 ± 0.46 *	5.35 ± 0.31 *†	4.76 ± 0.50 *†§
OGTT-G 60′ (mmol/L)	6.90 ± 1.49	12.02 ± 1.52 *	8.25 ± 1.42 *†	10.54 ± 1.08 *†§
OGTT-G 120′ (mmol/L)	5.64 ± 1.12	8.69 ± 1.33 *	5.97 ± 0.98 †	8.29 ± 1.49 *§
Fasting insulin, early pregnancy (µU/mL)	7.5 (5.3–10.7)	15.7 (14.3–27.5) *	11.6 (7.3–16.9) *†	9.8 (6.6–12.8) *†
Fasting C-Peptide, early pregnancy (ng/mL)	1.5 (1.2–1.9)	2.8 (2.3–3.5) *	1.9 (1.6–2.5) *†	1.8 (1.4–2.3) *†
QUICKI, early pregnancy (dimensionless) × 10^2^	36.2 ± 3.4	30.9 ± 1.8 *	33.9 ± 3.1 *†	34.9 ± 3.2 *†
TyGIS, early pregnancy (mg kg^−1^ min^−1^)	6.6 (5.6–7.4)	2.8 (1.6–4.0) *	5.3 (4.0–6.4) *†	5.8 (4.8–6.8) *†
IGI_F_, early pregnancy (ng/mg)	2.04 ± 0.75	3.19 ± 0.72 *	2.45 ± 0.82 *†	2.28 ± 0.73 *†
DI_ORAL_, early pregnancy (ng mg^−1^ (µU/mL)^−1^) × 10^2^	24.7 (20.6–30.8)	17.4 (13.2–19.0) *	21.1 (17.3–25.7) *†	22.8 (19.4–27.3) †

Data are mean ± SD or median (IQR) and count (%) for women remaining normal glucose tolerant (NGT) vs. patients developing gestational diabetes (GDM) categorized into clusters (CL 1–3). BMI, body mass index; FPG, fasting plasma glucose; HbA1c, glycated hemoglobin; QUICKI, quantitative insulin sensitivity check index; TyGIS, triglyceride-glucose insulin sensitivity index; IGI_F_, insulinogenic index; DI_ORAL_, disposition index; OGTT-G, oral glucose tolerance test—glucose levels. * *p* < 0.05 vs. NGT. † *p* < 0.05 vs. CL1. § *p* < 0.05 vs. CL2.

## Data Availability

The datasets used and/or analyzed during the current study are not publicly available due to ethical and privacy reasons, but are available from the corresponding author upon reasonable request.

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
