# Peer review of "Gestational Diabetes Mellitus Subtypes Derived by Clustering Analysis Show Heterogeneity in Glucometabolic Parameters Already at Early Pregnancy"

_nutrients, 2025, doi:10.3390/nu17203252_

Round 1
Reviewer 1 Report
Comments and Suggestions for Authors
The authors investigate the heterogeneity of gestational diabetes mellitus (GDM) using cluster analysis to verify established subtypes in an independent prospective study population. The authors demonstrate that glucometabolic differences emerge during early pregnancy, and these differences match specific clinical and metabolic risk patterns. The research provides timely results through sound methodology and clinical value because it demonstrates how early patient stratification enables better preventive and therapeutic strategies.
The research benefits from three major strengths, which include its substantial participant number, detailed participant characterization, and its combination of established and innovative tests for insulin sensitivity and beta-cell function. The research confirms and expands upon an existing cluster model, which enhances the reliability of this classification system. The study demonstrates that metabolic variations between clusters emerge before the middle of pregnancy, thus supporting individualized GDM treatment strategies.
The research contains specific points that need additional clarification. The small number of women in Cluster 1 reduces statistical power to evaluate pregnancy outcomes, including pharmacological treatment requirements and the occurrence of large-for-gestational-age infants. The discussion and conclusion should include a stronger emphasis on this study's limitations, as it prevents readers from drawing overly broad conclusions. The single food frequency questionnaire used to evaluate dietary habits during baseline measurement has limited ability to show how eating patterns change across pregnancy; thus, the authors need to provide additional details about this method. The statistical analysis performed Tukey’s HSD tests for subgroup comparisons but failed to implement multiple testing adjustments. The researchers should emphasize the exploratory nature of their analysis while also acknowledging the potential for increased type I error rates. The authors need to demonstrate that cluster-based risk stratification shows promise for intervention; however, they must also emphasize that actual clinical outcome improvements require interventional trials.
Author Response
Response to Reviewer 1
Dear Reviewer 1,
We sincerely thank Reviewer 1 for the thoughtful and constructive review of our manuscript. We greatly value the recognition of our study’s strengths and appreciate the time and effort dedicated to providing detailed feedback. The comments and suggestions were highly valuable and have helped us to revise and improve the manuscript.
Comment 1: The authors investigate the heterogeneity of gestational diabetes mellitus (GDM) using cluster analysis to verify established subtypes in an independent prospective study population. The authors demonstrate that glucometabolic differences emerge during early pregnancy, and these differences match specific clinical and metabolic risk patterns. The research provides timely results through sound methodology and clinical value because it demonstrates how early patient stratification enables better preventive and therapeutic strategies.
The research benefits from three major strengths, which include its substantial participant number, detailed participant characterization, and its combination of established and innovative tests for insulin sensitivity and beta-cell function. The research confirms and expands upon an existing cluster model, which enhances the reliability of this classification system. The study demonstrates that metabolic variations between clusters emerge before the middle of pregnancy, thus supporting individualized GDM treatment strategies.
Response 1: We thank the Reviewer for the positive and encouraging comment and the recognition of our study’s methodological and clinical strengths.
Comment 2: The research contains specific points that need additional clarification. The small number of women in Cluster 1 reduces statistical power to evaluate pregnancy outcomes, including pharmacological treatment requirements and the occurrence of large-for-gestational-age infants. The discussion and conclusion should include a stronger emphasis on this study's limitations, as it prevents readers from drawing overly broad conclusions. The single food frequency questionnaire used to evaluate dietary habits during baseline measurement has limited ability to show how eating patterns change across pregnancy; thus, the authors need to provide additional details about this method. The statistical analysis performed Tukey’s HSD tests for subgroup comparisons but failed to implement multiple testing adjustments. The researchers should emphasize the exploratory nature of their analysis while also acknowledging the potential for increased type I error rates. The authors need to demonstrate that cluster-based risk stratification shows promise for intervention; however, they must also emphasize that actual clinical outcome improvements require interventional trials.
Response 2: We thank the Reviewer for these constructive remarks. We agree that the limited sample size in Cluster 1 restricts the interpretation of pregnancy outcomes, and we have now more clearly emphasized this limitation in the Discussion and Conclusion sections (page 9, lines 261-264; page 10 lines 313-316, 338-339). Regarding dietary assessment, we explicitly noted its limitations for capturing longitudinal changes during pregnancy (page 10, lines 320-323). We also acknowledge that our subgroup analyses were exploratory, and we have clarified the potential risk of type 1 error due to multiple testing while stressing the hypothesis-generating nature of our findings (pages 3-4, lines 138-139; page 10, lines 323-325). Finally, we have revised the Discussion to underline that cluster-based risk stratification is promising, but that improvements in clinical outcomes require confirmation in dedicated interventional trials (page10, lines 326-330).
Concluding Remarks
We would like to thank Reviewer 1 once again for the constructive and helpful feedback. We believe that the manuscript has been significantly improved as a result of these valuable comments. All points raised by Reviewer 1 have been carefully addressed and the text revised accordingly. We hope that the revised version meets the expectations of Reviewer 1 as well as those of the editorial board, and we respectfully submit it for your consideration.
Sincerely,
Christian Göbl
On behalf of all authors
Reviewer 2 Report
Comments and Suggestions for Authors
Review of the manuscript entitled „Gestational Diabetes Mellitus Subtypes derived by Clustering Analysis show Heterogeneity in Glucometabolic Parameters already at Early Pregnancy” (Manuscript ID: nutrients-3876724)
The introduction clearly shows that gestational diabetes (GDM) is not a single condition, but rather a mixture of different patterns. It also shows that women often need care that is tailored to their specific situation. It then transitions smoothly to the concept of categorizing patients into subgroups or clusters. The study's aim is straightforward: to confirm the existence of three previously described GDM subtypes in a new group of women and to compare them with women who remained free of diabetes during pregnancy. The methods section carefully explains who was included, what tests were performed, and how the groups were formed. This makes the work transparent and trustworthy. The results naturally follow these steps and are presented in tables and figures that highlight the differences between the groups. The box plots and comparison with a simpler classification method are especially useful in demonstrating the strength of the cluster model. However, the small number of women in cluster 1 weakens the certainty of some findings. Additionally, the text sometimes repeats information and does not emphasize results that showed no difference enough. The discussion effectively ties back to the study's objectives, situating the findings alongside earlier research and demonstrating that this straightforward model, based on routine tests, can rival more intricate, physiology-based approaches. It emphasizes that cluster 1 carries the highest risk and introduces possible genetic influences and the role of early lifestyle changes. This practical perspective is a clear strength. However, the study's limitations, such as its modest sample size, single-center setting, and restricted diversity, should be more openly highlighted. Overall, the manuscript offers a coherent and well-structured narrative with strong data and relevant comparisons. However, it would benefit from tighter wording and a more balanced discussion of its limitations.
Comments:
- The introduction is too brief. While it acknowledges gestational diabetes (GDM) as a heterogeneous condition, it does not sufficiently draw on the extensive international literature describing different phenotypes and their links to outcomes. A stronger review of existing studies in this section would better demonstrate how the present work fits into ongoing research. Relevant references include Mahajan et al., who used genetic and clustering methods to demonstrate the heterogeneity of type 2 diabetes. While not directly related to GDM, their findings provide a conceptual framework by emphasizing that diabetes is not a single disease and that categorizing patients into subgroups is crucial for improved risk prediction and personalized care. Lowe and the HAPO studies have also highlighted diverse pathways leading to GDM and the graded impact of hyperglycemia on pregnancy outcomes. Catalano has emphasized the central role of maternal obesity and insulin resistance, and Farrar has demonstrated that different OGTT profiles pose distinct risks to mothers and children. Together, these studies and the genetic insights from Mahajan strengthen the idea that breaking down heterogeneous diabetes populations into meaningful subgroups is both clinically useful and necessary.
References:
- Mahajan A, Taliun D, Thurner M, Robertson NR, Torres JM, Rayner NW, et al. Fine-mapping type 2 diabetes loci to single-variant resolution using high-density imputation and islet-specific epigenome maps. Diabetologia. 2018
- HAPO Study Cooperative Research Group, Metzger BE, Lowe LP, Dyer AR, Trimble ER, Chaovarindr U, et al. Hyperglycemia and adverse pregnancy outcomes. N Engl J Med. 2008
- Lowe WL Jr, Scholtens DM, Lowe LP, Kuang A, Nodzenski M, Talbot O, et al. Association of gestational diabetes with maternal disorders of glucose metabolism and childhood adiposity. Diabetes Care. 2019
- Catalano PM, Shankar K. Obesity and pregnancy: mechanisms of short term and long term adverse consequences for mother and child. BMJ. 2017
- Farrar D, Simmonds M, Bryant M, Lawlor DA, Dunne F, Tuffnell D, et al. Hyperglycaemia and risk of adverse perinatal outcomes: systematic review and meta-analysis. BMJ. 2016
- To emphasize the topic's current relevance, it would also be advisable to include more recent literature.
References:
- Gong Y, et al. Heterogeneity of Gestational Diabetes and Risk for Adverse Pregnancy Outcomes: A Cohort Study. The Journal of Clinical Endocrinology & Metabolism. 2025
- Liu Y, et al. Heterogeneity of insulin resistance and beta cell dysfunction in gestational diabetes mellitus: different impacts on perinatal outcomes. Journal of Translational Medicine. 2018
- Athanasiadou KI, et al. Gestational diabetes mellitus subtypes according to oral glucose tolerance test and pregnancy outcomes. Endocrine. 2025
- The description of insulin treatment types (long- versus short-acting) seems excessive, especially since the main focus of the paper is on phenotypic differences. This risks diverting attention from the core research question. The introduction would benefit from a clearer focus on heterogeneity and the clinical relevance of clustering.
- The presentation of laboratory indices, such as IGI and DI, is fragmented and difficult to follow. It would be clearer if they were laid out in a more logical sequence.
- In the statistical section, stating that "no further adjustment for multiple testing" was applied weakens the reliability of the findings and should be acknowledged as a limitation.
- The Methods section provides a thorough technical description of the clustering procedure, but the Introduction does not mention that the study also aimed to compare clusters against a simple categorization using kappa statistics. This should be corrected to improve consistency.
- The justification for the sample size is weak ("no post hoc power calculation"), and, since this is a secondary analysis, the limited statistical power should be recognized more explicitly. The adequacy of the sample size for this study's aims should be explained in greater detail and reiterated in the discussion because it directly affects the generalizability of the conclusions.
- Cluster 1 is very small (n = 15, or 7.7%), which substantially increases the uncertainty of the results. While the Methods section acknowledges this, the Results section should also emphasize the limitations of statistical power, particularly when analyzing binary outcomes such as LGA or insulin treatment.
- Important clinical outcomes, including pregnancy complications and LGA, did not differ between clusters. This key negative finding deserves more emphasis, particularly since the introduction suggests that the clusters might predict such outcomes.
- Several data points are presented twice: first in the text, such as BMI and OGTT differences, and again in the tables. The text would be clearer if it only highlighted the main findings and left the detailed values in the tables.
- The fair-to-moderate agreement (κ = 0.44) is an important result, yet the results section does not explain its clinical significance. Although a fuller interpretation belongs in the discussion section, the results should at least emphasize that this does not represent strong concordance.
- The Methods section mentions that no correction for multiple testing was applied except for Tukey HSD, yet the Results section presents many p-values below 0.05. These should be handled more cautiously with consistent acknowledgment that the analyses are exploratory.
- The results clearly show that there were no significant differences in LGA rates or the need for medication; however, the discussion only briefly mentions this. It should be emphasized that the clusters did not show predictive value for clinical outcomes in this sample.
- Certain points, such as the combination of insulin resistance and elevated triglycerides in Cluster 1, are repeated several times almost verbatim and should be condensed for clarity.
- The section on genetic background is interesting, but it is presented too briefly and mainly as a list of references. It is not fully integrated into the main narrative. Adding a separate paragraph on "future directions in genetic heterogeneity" would strengthen this section.
- The discussion of lifestyle interventions is rather general and does not directly link to the clusters (for example, the limitations of FFQs). It would be more useful to emphasize that cluster 1 is the group most likely to benefit from early intervention.
- Additional limitations of the study:
- Single-center design, which raises questions about how widely the findings can be generalized, particularly in an international context.
- Although ethnicity was recorded, the sample appears to consist mainly of European women, leaving other groups underrepresented.
- Key metabolic parameters, such as insulin, C-peptide, and lipid levels, were measured only once; therefore, they may not reflect changes throughout pregnancy.
- Dietary data were collected through a single self-reported questionnaire, which is prone to bias and may overlook shifts in eating habits over time.
- The analysis focused primarily on metabolic markers and had limited statistical power to detect differences in perinatal outcomes.
- The large number of p-values reported with minimal adjustment for multiple comparisons increases the risk of false-positive findings.
Author Response
Response to Reviewer 2
Dear Reviewer 2,
We sincerely thank Reviewer 2 for the thoughtful and constructive review of our manuscript. We greatly value the recognition of our study’s strengths and appreciate the time and effort dedicated to providing detailed feedback. The comments and suggestions were highly valuable and have helped us to revise and improve the manuscript.
Comment 1: The introduction clearly shows that gestational diabetes (GDM) is not a single condition, but rather a mixture of different patterns. It also shows that women often need care that is tailored to their specific situation. It then transitions smoothly to the concept of categorizing patients into subgroups or clusters. The study's aim is straightforward: to confirm the existence of three previously described GDM subtypes in a new group of women and to compare them with women who remained free of diabetes during pregnancy. The methods section carefully explains who was included, what tests were performed, and how the groups were formed. This makes the work transparent and trustworthy. The results naturally follow these steps and are presented in tables and figures that highlight the differences between the groups. The box plots and comparison with a simpler classification method are especially useful in demonstrating the strength of the cluster model. However, the small number of women in cluster 1 weakens the certainty of some findings. Additionally, the text sometimes repeats information and does not emphasize results that showed no difference enough. The discussion effectively ties back to the study's objectives, situating the findings alongside earlier research and demonstrating that this straightforward model, based on routine tests, can rival more intricate, physiology-based approaches. It emphasizes that cluster 1 carries the highest risk and introduces possible genetic influences and the role of early lifestyle changes. This practical perspective is a clear strength. However, the study's limitations, such as its modest sample size, single-center setting, and restricted diversity, should be more openly highlighted. Overall, the manuscript offers a coherent and well-structured narrative with strong data and relevant comparisons. However, it would benefit from tighter wording and a more balanced discussion of its limitations.
Response 1: We thank the Reviewer for the thorough review and constructive comments. In response, we have revised the manuscript in order to reduce repetitions and clarified results that showed no statistically significant differences. We also revised the Discussion to place stronger emphasis on the limitations, including the small number of women in Cluster 1, the single-center setting, and the restricted diversity of the cohort (page 10, lines 311-330).
Comment 2: The introduction is too brief. While it acknowledges gestational diabetes (GDM) as a heterogeneous condition, it does not sufficiently draw on the extensive international literature describing different phenotypes and their links to outcomes. A stronger review of existing studies in this section would better demonstrate how the present work fits into ongoing research. Relevant references include Mahajan et al., who used genetic and clustering methods to demonstrate the heterogeneity of type 2 diabetes. While not directly related to GDM, their findings provide a conceptual framework by emphasizing that diabetes is not a single disease and that categorizing patients into subgroups is crucial for improved risk prediction and personalized care. Lowe and the HAPO studies have also highlighted diverse pathways leading to GDM and the graded impact of hyperglycemia on pregnancy outcomes. Catalano has emphasized the central role of maternal obesity and insulin resistance, and Farrar has demonstrated that different OGTT profiles pose distinct risks to mothers and children. Together, these studies and the genetic insights from Mahajan strengthen the idea that breaking down heterogeneous diabetes populations into meaningful subgroups is both clinically useful and necessary.
References:
- Mahajan A, Taliun D, Thurner M, Robertson NR, Torres JM, Rayner NW, et al. Fine-mapping type 2 diabetes loci to single-variant resolution using high-density imputation and islet-specific epigenome maps. Diabetologia. 2018
- HAPO Study Cooperative Research Group, Metzger BE, Lowe LP, Dyer AR, Trimble ER, Chaovarindr U, et al. Hyperglycemia and adverse pregnancy outcomes. N Engl J Med. 2008
- Lowe WL Jr, Scholtens DM, Lowe LP, Kuang A, Nodzenski M, Talbot O, et al. Association of gestational diabetes with maternal disorders of glucose metabolism and childhood adiposity. Diabetes Care. 2019
- Catalano PM, Shankar K. Obesity and pregnancy: mechanisms of short term and long term adverse consequences for mother and child. BMJ. 2017
- Farrar D, Simmonds M, Bryant M, Lawlor DA, Dunne F, Tuffnell D, et al. Hyperglycaemia and risk of adverse perinatal outcomes: systematic review and meta-analysis. BMJ. 2016
Response 2: We thank the Reviewer for this valuable suggestion. We have substantially expanded the Introduction to place our study more clearly within the context of existing international literature. In particular, we now reference and discuss the suggested literature (page 2, lines 43-46, 52-58).
Comment 3: To emphasize the topic's current relevance, it would also be advisable to include more recent literature.
References:
- Gong Y, et al. Heterogeneity of Gestational Diabetes and Risk for Adverse Pregnancy Outcomes: A Cohort Study. The Journal of Clinical Endocrinology & Metabolism. 2025
- Liu Y, et al. Heterogeneity of insulin resistance and beta cell dysfunction in gestational diabetes mellitus: different impacts on perinatal outcomes. Journal of Translational Medicine. 2018
- Athanasiadou KI, et al. Gestational diabetes mellitus subtypes according to oral glucose tolerance test and pregnancy outcomes. Endocrine. 2025
Response 3: We thank the Reviewer for providing additional literature which was now included in the revised version of the manuscript (page 2, lines 66-67).
Comment 4: The description of insulin treatment types (long- versus short-acting) seems excessive, especially since the main focus of the paper is on phenotypic differences. This risks diverting attention from the core research question. The introduction would benefit from a clearer focus on heterogeneity and the clinical relevance of clustering.
Response 4: We revised the Introduction to emphasize heterogeneity of GDM and the clinical relevance of clustering as suggested by the Reviewer (page 2, lines 43-46, 48-49).
Comment 5: The presentation of laboratory indices, such as IGI and DI, is fragmented and difficult to follow. It would be clearer if they were laid out in a more logical sequence.
Response 5: The section has been revised to present IGI and DI in a clearer, more logical sequence. Also, in Table 1 we shifted upwards the OGTT glucose values that had been erroneously reported after the insulin sensitivity and β-cell function indices, which are now reported at the bottom of the table, in the right order and position (page 3, lines 105-112; Table 1).
Comment 6: In the statistical section, stating that "no further adjustment for multiple testing" was applied weakens the reliability of the findings and should be acknowledged as a limitation.
Response 6: This limitation is now mentioned and addressed in the Discussion section (page 10, lines 323-325).
Comment 7: The Methods section provides a thorough technical description of the clustering procedure, but the Introduction does not mention that the study also aimed to compare clusters against a simple categorization using kappa statistics. This should be corrected to improve consistency.
Response: 7: We addressed this point at the end of the Introduction in accordance with the comment of the Reviewer (page 2, lines 74-76).
Comment 8: The justification for the sample size is weak ("no post hoc power calculation"), and, since this is a secondary analysis, the limited statistical power should be recognized more explicitly. The adequacy of the sample size for this study's aims should be explained in greater detail and reiterated in the discussion because it directly affects the generalizability of the conclusions.
Response 8: Post hoc power calculations can be criticized (PMID: 32843199). However, we have now carefully addressed the mentioned limitation of modest sample size, especially in cluster 1. The aim of the study is to generate new hypotheses – in this regard we achieved our goal as we found impaired glucose metabolism was present already in early pregnancy of women categorized as GDM cluster 1 but we have limited power for binary outcomes; such as insulin treatment or LGA rates. This is now addressed in more detail at the end of the Discussion section (page 10, lines 311-316).
Comment 9: Cluster 1 is very small (n = 15, or 7.7%), which substantially increases the uncertainty of the results. While the Methods section acknowledges this, the Results section should also emphasize the limitations of statistical power, particularly when analyzing binary outcomes such as LGA or insulin treatment.
Response 9: We agree with the Reviewer and addressed the limited sample size for Cluster 1 as a limitation of this study – especially considering binary obstetric outcomes as mentioned above (page 10, lines 313-316).
Comment 10: Important clinical outcomes, including pregnancy complications and LGA, did not differ between clusters. This key negative finding deserves more emphasis, particularly since the introduction suggests that the clusters might predict such outcomes.
Response 10: We thank the Reviewer for this important comment. We have revised the Discussion to emphasize more clearly that pregnancy complications and LGA did not differ between clusters, highlighting this as a negative finding in contrast to the initial hypothesis. However, we now also indicate that this lack of difference may in part reflect the limited sample size in Cluster 1 (page 8, lines 216-218; page 10, lines 313-316).
Comment 11: Several data points are presented twice: first in the text, such as BMI and OGTT differences, and again in the tables. The text would be clearer if it only highlighted the main findings and left the detailed values in the tables.
Response 11: The Results were revised and shortened wherever possible (page 4, lines 155-157) .
Comment 12: The fair-to-moderate agreement (κ = 0.44) is an important result, yet the results section does not explain its clinical significance. Although a fuller interpretation belongs in the discussion section, the results should at least emphasize that this does not represent strong concordance.
Response 12: We thank the Reviewer for this valuable remark. We have revised the Results and Discussion section to clarify that the observed agreement is not indicative of strong agreement (page 7, lines 201-202; page 9, lines 273-274).
Comment 13: The Methods section mentions that no correction for multiple testing was applied except for Tukey HSD, yet the Results section presents many p-values below 0.05. These should be handled more cautiously with consistent acknowledgment that the analyses are exploratory.
Response 13: We have now explicitly addressed the exploratory character of this study aiming to generate new hypotheses which need to be addressed in future studies (page 3-4, lines 138-139).
Comment 14: The results clearly show that there were no significant differences in LGA rates or the need for medication; however, the discussion only briefly mentions this. It should be emphasized that the clusters did not show predictive value for clinical outcomes in this sample.
Response 14: We thank the Reviewer for this important point. We have revised the Discussion to explicitly emphasize that, in this cohort, the clusters did not predict clinical outcomes such as LGA rates or the need for pharmacological treatment (page 8, lines 216-218; page 9, lines 261-264; page 10 lines 313-316).
Comment 15: Certain points, such as the combination of insulin resistance and elevated triglycerides in Cluster 1, are repeated several times almost verbatim and should be condensed for clarity.
Response 15: We thank the Reviewer for this observation and have revised the manuscript to condense repetitive statements.
Comment 16: The section on genetic background is interesting, but it is presented too briefly and mainly as a list of references. It is not fully integrated into the main narrative. Adding a separate paragraph on "future directions in genetic heterogeneity" would strengthen this section.
Response 16: We thank the Reviewer for this helpful suggestion. We have revised the section on genetic background to improve integration into the main narrative and extended the paragraph to highlight its potential relevance for GDM research (pages 9-10, lines 297-308).
Comment 17: The discussion of lifestyle interventions is rather general and does not directly link to the clusters (for example, the limitations of FFQs). It would be more useful to emphasize that cluster 1 is the group most likely to benefit from early intervention.
Response 17: We agree that early intervention is specifically necessary for patients who develop GDM cluster 1. This is now addressed with more detail in the Discussion (page 9, lines 276-280, line 286).
Comment 18: Additional limitations of the study:
- Single-center design, which raises questions about how widely the findings can be generalized, particularly in an international context.
- Although ethnicity was recorded, the sample appears to consist mainly of European women, leaving other groups underrepresented.
- Key metabolic parameters, such as insulin, C-peptide, and lipid levels, were measured only once; therefore, they may not reflect changes throughout pregnancy.
- Dietary data were collected through a single self-reported questionnaire, which is prone to bias and may overlook shifts in eating habits over time.
- The analysis focused primarily on metabolic markers and had limited statistical power to detect differences in perinatal outcomes.
- The large number of p-values reported with minimal adjustment for multiple comparisons increases the risk of false-positive findings.
Response 18: The limitations are now addressed in the revised version of the manuscript (page 9, lines 261-264; page 10, lines 311-325).
Concluding Remarks
We would like to thank Reviewer 2 once again for the constructive and helpful feedback. We believe that the manuscript has been significantly improved as a result of these valuable comments. All points raised by Reviewer 2 have been carefully addressed and the text revised accordingly. We hope that the revised version meets the expectations of Reviewer 2 as well as those of the editorial board, and we respectfully submit it for your consideration.
Sincerely,
Christian Göbl
On behalf of all authors
Round 2
Reviewer 1 Report
Comments and Suggestions for Authors
Accept in present form
Reviewer 2 Report
Comments and Suggestions for Authors
I thank the authors and the editor for their thorough, excellent work, I accept the responses and recommend publishing the manuscript!